# Designing Deep Learning Programs with Large Language Models

## Abstract

The process of utilizing deep neural architectures to solve tasks differs significantly from conventional programming due to its complexity and the need for specialized knowledge. While code generation technologies have made substantial progress, their application in deep learning programs requires a distinct approach. Although previous research has shown that large language model agents perform well in areas such as data science, neural architecture search, and hyperparameter tuning, the task of proposing and refining deep neural architectures at a high level remains largely unexplored. Current methods for automating the synthesis of deep learning programs often rely on basic code templates or API calls, which restrict the solution space to predefined architectures. In this paper, we aim to bridge the gap between traditional code generation and deep learning program synthesis by introducing the task of Deep Learning Program Design (DLPD), a task of designing an effective deep learning program for the task, along with appropriate architectures and techniques. We propose Deep Ones, a comprehensive solution for DLPD. Our solution includes a large-scale dataset and a lightweight benchmark specifically designed for DLPD. On our benchmark, Llama-3.1 8B, fine-tuned on our dataset, demonstrates better architecture suggestion capability than GPT-4o and better performance than Claude-3.5-Sonnet, showcasing that Deep Ones effectively addresses the challenge of DLPD. Deep Ones will be publicly available, including the dataset, benchmark, codes, and model weights.

## 1 Introduction

Program synthesis, the process of automatically generating software from high-level specifications, has gained significant attention due to its practicality. With recent advancements in deep learning and large language models (LLMs), many studies have proposed models capable of generating source code through pre-training (Wang et al., 2021; Di et al., 2023; Li et al., 2023), additional fine-tuning (Austin et al., 2021; Chen et al., 2021), evolutionary algorithm (Luo et al., 2023), or reinforcement learning (Le et al., 2022; Shojaee et al., 2023). These methods have shown strong performance in code generation benchmarks such as APPS (Hendrycks et al., 2021), MBPP (Austin et al., 2021), and HumanEval (Chen et al., 2021). However, since these benchmarks primarily address relatively simple code generation tasks, recent efforts have shifted toward generating more complex programs, such as competition-level codes (Li et al., 2022b; Ridnik et al., 2024), data science programs (Lai et al., 2023; Chandel et al., 2022), class-level codes (Du et al., 2023), and repository-level codes (Zhang et al., 2023a).

Nevertheless, such approaches have not been actively investigated within the domain of deep learning programs, which involves generating an executable code that utilizes a deep neural architecture. This is due to several open challenges: complex code structures, sophisticated environment configurations, and poorly-defined evaluation standards.

As a result, most prior research in this area has focused on problem-solving API usage (Shen et al., 2024; Patil et al., 2023; Ge et al., 2024; Gao et al., 2023; Liang et al., 2023), automated architectural modification of a pre-defined code or hyperparameter tuning (Huang et al., 2023; Zhang et al., 2023c; Liu et al., 2024), and neural architecture search (Elsken et al., 2019) along with AutoML (He et al., 2021). However, these approaches often operate within a limited solution space, relying on slight modifications of predefined architectures. While using a base model guarantees performance

with relatively safe execution, it leaves little room for architectural designs or technical enhancement typically made by human researchers, which may boost performance much greater than layer changing or hyperparameter tuning.

To bridge the gap between low-level code generation technologies and high-level deep learning architecture usage, we propose the task of Deep Learning Program Design (DLPD), a task of designing an effective deep learning program for the task utilizing appropriate architectures and techniques. Additionally, to cope with the aforementioned challenges, we present **DeepOnes**, a comprehensive solution for DLPD. DeepOnes consists of a large-scale dataset, a multiple-choice QA benchmark, and a lightweight benchmark specifically tailored for evaluating the program design capabilities. We coin these components as **DeepData**, **DeepQA**, and **DeepBench**, respectively.

For effective program design, we assume that large language models must possess extensive knowledge of various architectures and techniques for flexible improvement. However, to the best of our knowledge, no existing dataset comprehensively covers deep neural architectures and their associated techniques. To address this gap, we introduce DeepData, a novel dataset comprising rich information extracted from arXiv papers and corresponding implementations on GitHub. Inspired by biomolecular knowledge tasks from Mol-Instructions (Fang et al., 2023), we organize the data into various categories, including description generation, combination prediction, property prediction, reasoning, mathematical expression, name guessing, and more. We further process the data for DLPD, by articulating the tasks of requirement-based model suggestion, property-based improvement suggestion, and hyperparameter prediction. For a 0.01% subset of these papers, we also collect multiple-choice questions to evaluate the knowledge of current LLMs in the domain of deep learning techniques.

To further evaluate models and establish benchmarks, we present **DeepBench**, a benchmark that consists of 10 deep learning tasks collected from Papers with Code (PWC)[1], spanning text, image, and audio modalities. **DeepBench** evaluates a model's program designing ability by generating a fully executable deep learning program based on the given design. The benchmark utilizes a **generate-then-improve** framework to evaluate if LLM can truly make an appropriate architectural or technical improvement, not merely repeating the existing solution.

In summary, our contributions are as follows:

- To bridge the gap between low-level code generation technologies and high-level deep learning architecture usage, we propose the task of deep learning program design and DeepOnes, a comprehensive solution to this task.
- We introduce DeepData, the first dataset tailored for the task of DLPD. This includes synthetic data created from research papers and corresponding GitHub repositories, augmented using LLMs.
- From a small portion of research papers used for DeepData, we collect DeepQA, the first multiple choice question-answering benchmark for the topics on artificial intelligence. We evaluate several open-source and closed-source LLMs on DeepQA to analyze the amount of knowledge they possess and we show that the model trained on DeepData outperforms all the other baselines.
- We create the DeepBench benchmark, comprised of 10 general deep learning tasks across several modalities, collected from Papers With Code. This includes the pipeline that synthesizes a fully-executable deep learning program from a natural language task description.
- We release all the datasets, codes, model weights, and benchmark so that the open-source community can make improvements in the field of DLPD.

## 2 RELATED WORK

### 2.1 PROGRAM SYNTHESIS

Program synthesis is defined as automating the software development process from declarative specification (Kreitz, 1998). Earlier work mostly focused on utilizing theorem-proving techniques (Green, 1981; Waldinger & Lee, 1969; Stark & Ireland, 1999). Since the emergence of

---

[1]https://paperswithcode.com/

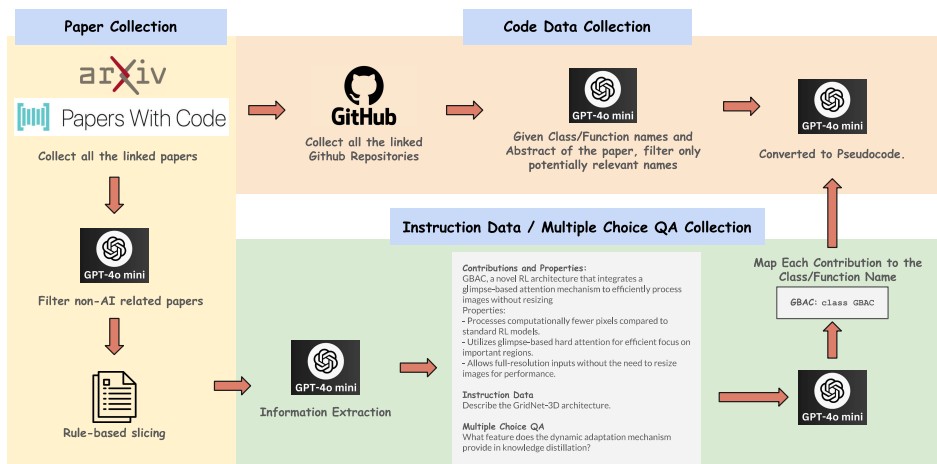

Figure 1: **Pipeline for generating DeepData and DeepQA**

LLMs with strong language capabilities (Devlin et al., 2016; Brown et al., 2020; Raffel et al., 2020), there have been increased interest in understanding and generating source code using LLMs (Feng et al., 2020; Lu et al., 2021; Ahmad et al., 2021; Wang et al., 2021; Chen et al., 2021; Di et al., 2023; Li et al., 2023; Luo et al., 2023). Recently, LLMs have demonstrated impressive performance not only on the function-level, but also on the class-level (Du et al., 2023), the repository-level (Zhang et al., 2023a), or on the Jupyter Notebook-level (Chandel et al., 2022). Nonetheless, the technologies are rarely applied for deep learning codes due to the huge domain difference and the lack of test cases.

## 2.2 LLMs for Automated Machine Learning

Recently, since the introduction of high-performing LLMs, there have been studies exploring the automation of machine learning pipeline using them. AutoML-GPT (Zhang et al., 2023c) and AgentHPO (Liu et al., 2024) provides LLMs with rich information through model cards and data cards, allowing LLMs to predict better hyperparameters even for the unseen tasks. MLCopilot (Zhang et al., 2023b) designs a two-stage strategy of an information-gathering offline stage and retrieval-augmented task solving online stage. MLAgentBench (Huang et al., 2023) proposes a benchmark to evaluate an LLM agent's ability to improve a starter code's performance on various tasks, and employs an LLM agent with pre-action thoughts such as reflection, research plan, status, and fact check. On the other hand, some other works focus on integrating LLMs with a set of APIs as tools. HuggingGPT (Shen et al., 2024), Gorilla (Patil et al., 2023), and OpenAGI (Ge et al., 2024) take advantage of famous API storage, such as Huggingface, Tensorflow-hub, Pytorch-hub, Github, or Langchain, while AssistGPT (Gao et al., 2023) focuses more on utilizing predefined tools. Recent researches even attempt to automate the data-driven discovery (Gu et al., 2024; Majumder et al., 2024; Guo et al., 2024), but they focus on data science analysis rather than the deep learning architecture itself. No study on automated machine learning focuses on designing or modifying the architecture itself, mostly taking advantage of retrieval or API.

## 3 DeepData: Dataset for Deep Neural Architecture Desgin

We introduce DeepData, a novel dataset specifically designed for DLPD. DeepData consists of 1,346,051 instruction data points and 2,080,274 DLPD data points, gathered from 325,301 research papers related to artificial intelligence. This dataset is derived from research papers and corresponding code implementations, which we collected from the Papers with Code (PWC) platform[2] and Github[3] repositories. PWC provides links to research papers on arXiv and associated Github repositories that implement the methods discussed in the papers. Our dataset includes research papers and

---

[2] https://paperswithcode.com/
[3] https://github.com/

**Description Generation**
Could you explain the Glimpse-Based Actor-Critic (GBAC) model?

**Description Generation**
What is the main function of TASD in natural language processing?

**Combination Prediction**
How can we combine the Dynamic Grained Encoder with other transformer frameworks?

**Combination Prediction**
Which of the following is a method to reduce localization errors in AUVs?

**Property Prediction**
What benefits does the hard attention mechanism provide in GBAC?

**Property Prediction**
What advantage does relaxed-LSS provide over traditional leverage score sampling?

**Reasoning**
Explain how the GBAC model can match the performance of PPO despite processing fewer pixels.

**Reasoning**
How does focal loss improve model performance in classification tasks?

**Mathematical Expression**
Describe the Proximal Policy Optimization (PPO) algorithm mathematically.

**Mathematical Expression**
What theoretical dimension bound is necessary for reliably estimating set intersections in VSAs?

**Name Guess**
Is there a technique that combines hard attention and reinforcement learning with efficient computation?

**Name Guess**
Which method does FewSOME utilize to prevent representational collapse?

**Miscellaneous**
What are potential applications of measuring game distance in Ludii? (Application Prediction)

**Miscellaneous**
What is the result of using the proposed relational network on the visual XOR task? (Score)

Figure 2: **Samples of DeepData and DeepQA** Data on the left side is instruction-style DeepData and data on the right side is multiple-choice question-style DeepQA. Full examples are available on the Appendix B.2

**Requirement-based Architecture Proposal**
What would be the model that integrates novel word orders based on dependency parsing for downstream nlp tasks under these constraints?
- gpu memory: Uses approximately 15GB of GPU memory for training.
- train time: Requires around 10 hours for training with optimizations.
- inference time: Achieves inference in less than 1 second per example.
=> One possible model is WordOrderOptimizer. It integrates novel word orders based on dependency parsing for downstream nlp tasks, and utilizes a combinatorial optimization framework to ensure effective representation of natural language structure.

**Property-based Architecture Improvement**
What would be the block that improves sensitivity to border features crucial for accurately detecting rotated objects?
=> One possible block is rotation-aware deformable convolution. It enhances feature extraction by adapting the convolution sampling points based on object orientation, and improves sensitivity to border features crucial for accurately detecting rotated objects.

**Hyperparameter Prediction**
What is a recommended learning rate for COMBO Architecture?
=> 0.002

Figure 3: **Samples of DLPD DeepData.** Three categories of DLPD-tuning data are included in DeepData: requirement-based architecture proposal, property-based architecture imporvement, and hyperparameter prediction.

code repositories from January 2013 to July 2024. Due to the limitation in space, we only showcase brief samples of data in the Figure 2. Full examples can be found in the Appendix B.2, while full prompts used for processing the dataset can be found in the supplementary material. We mainly used GPT-4o-mini (Achiam et al., 2023) to extract and synthesize the data. The pipeline for dataset synthesis is visualized on the Figure 1.

## 3.1 RESEARCH PAPERS

Although PWC is a platform that includes a large amount of AI-related papers, non-AI papers are also in the database. Thus, using GPT-4o-mini, we first extracted only the research papers relevant to AI-related technology by providing the abstract of a paper. Then, we parsed .tex files of the arXiv research papers using the unarXive (Saier et al., 2023) to retrieve clean text including the math-

ematical equations. Further using GPT-4o-mini, we have extracted (1) contributions, (2) paper's contribution represented as a diagram (3) properties of each node in a diagram (4) requirements such as GPU or time, (5) instruction data, and for small portion, (6) multiple-choice questions described in the section 3.4. Instruction data and multiple choice questions are categorized as several categories, including description generation, combination prediction, property prediction, reasoning, mathematical expressions, name guessing, and more. The distribution of each category can be found in the AppendixA.

## 3.2 GITHUB REPOSITORY

In addition to research papers, we have used the Github API to retrieve repositories linked from the papers. From each repository, we extracted only functions and classes, assuming that the contributions proposed in the papers are mostly implemented as functions or classes. To reduce excessive number of tokens, we further filtered the functions and classes using GPT-4o-mini to identify only classes and functions that are potentially relevant to the paper's abstract. We make a mapping between the extracted codes and each node in a diagram using GPT-4o-mini, For example, the node `<MODEL>CNN` may be mapped to the function `def CNN`. Finally, GPT-4o-mini converts the class or function into pseudocode, focusing on high-level functionality of it. This is because raw code snippets include a lot of noises, which does not relate to the main functionality and make a relatively small LLM hard to learn from it.

## 3.3 PREPROCESSING FOR ARCHITECTURE DESIGN

Using the data collected from research papers, we created an additional synthetic dataset tailored for DLPD. Neural architectures are often developed by combining, modifying, or replacing existing components based on their properties. For example, ResNet (He et al., 2016) improved the performance of CNNs by introducing residual connections.

Motivated by this idea, we categorized the program designing task into three subtasks: (1) proposing existing architectures based on requirements (e.g., GPU, time, or task), (2) modifying architectures based on component properties, and (3) selecting appropriate hyperparameters. The first dataset consists of requirement-model pairs, the second of component-property pairs (e.g., "residual connections improve performance and reduce overfitting"), and the third of model-hyperparameter pairs. The examples are available on the Figure 3.

As a result, DeepData includes two types of data pairs to fine-tune models. We first fine-tune the LLMs on instruction-style data to inject enough background knowledge on AI-related technologies. Then, we further fine-tune using DLPD-style data to train it to effectively perform program design and even generate corresponding pseudocode which guides the programmer model to implement it.

## 3.4 DEEPQA

On the process of extracting DeepData's instruction data, for 1% of the papers, we additionally synthesized multiple-choice questions based on the papers. Being consistent with instruction data, we collect the question categories of description generation, combination prediction, property prediction, reasoning, mathematical expressions, name guessing, and others. We have collected 8,851 multiple choice questions until 2023 December. On the Figure 4, we show that GPT-4o has already learned most of the AI-related knowledge, while Claude 3.5 Sonnet has a poor capability comparable to 8B open-source models. DeepLlama-8B, a Llama-3.1-8B trained on DeepData, outperforms all the other baselines, demonstrating that the model successfully learns AI-related knowledge from the dataset. The examples of the questions are on the Figure 2, while full examples are on Appendix B.2.

## 4 DEEPBENCH

In this section, we introduce a new benchmark DeepBench to evaluate our pipeline on the task of DLPD. This benchmark includes 10 popular tasks collected from PWC, ranging different modalities. We pair it with a relatively popular and small datasets for rapid evaluation and assign one metric for simplicity.

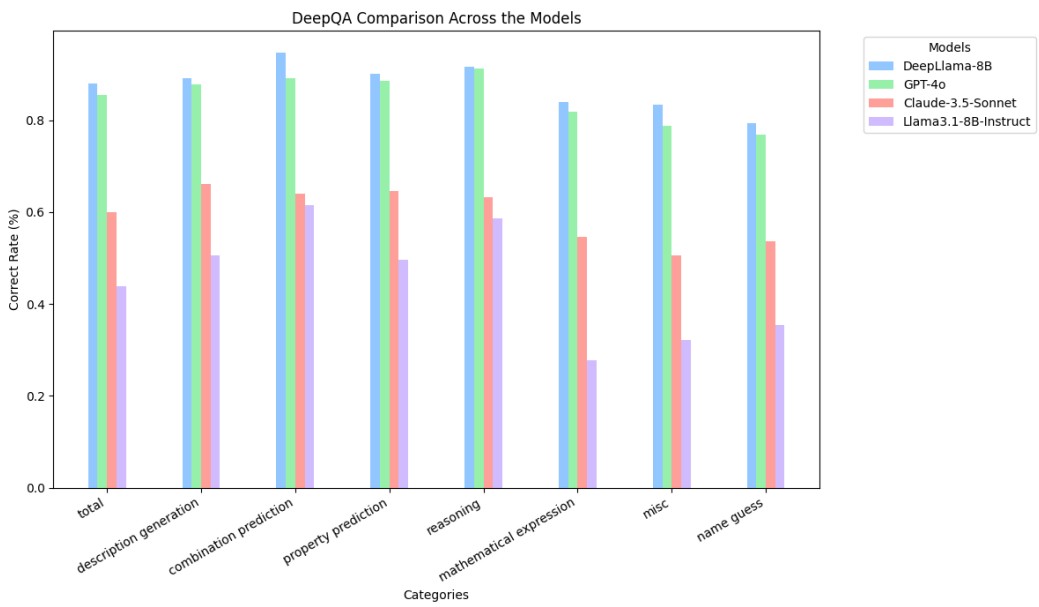

Figure 4: **Evaluation on DeepQA** While GPT-4o already possesses most of the knowledge, Claude 3.5 Sonnet and Llama3.1 8B-Instruct fails to solve the problems in many cases. DeepLlama, which is Llama 3.1 8B fine-tuned on our instruction dataset, shows the best scores in all the categories.

| Task | Modality | Dataset | Metric | PWC SOTA |
|---|---|---|---|---|
| image classification | Image | CIFAR-10 (Krizhevsky et al., 2009) | Accuracy (↑) | 99.61 (Bruno et al., 2022) |
| text-to-image generation | Text, Image | MS COCO$_{mini}$ (Lin et al., 2014) | FID (↓) | 3.22 (Yu et al., 2022) |
| image captioning | Image | MS COCO$_{mini}$ (Lin et al., 2014) | BLEU-4 (↑) | 46.5 (Li et al., 2022a) |
| object detection | Image | MS COCO$_{mini}$ (Lin et al., 2014) | Box Average Precision (↑) | 58.1 (Hou et al., 2024) |
| face recognition | Image | LFW (Huang et al., 2008) | Accuracy (↑) | 99.87 (Alansari et al., 2023) |
| question answering | Text | GLUE QNLI (Wang, 2018) | Accuracy (↑) | 99.2 (Lan, 2019) |
| sentiment classification | Text | GLUE SST2 (Wang, 2018) | Accuracy (↑) | 94.38 (Huang et al., 2020) |
| natural language inference | Text | GLUE MNLI (Wang, 2018) | Accuracy (↑) | 92.0 (Jiang et al., 2019) |
| recommendation system | Text | MovieLens-100K (Harper & Konstan, 2015) | RMSE (↓) | 0.887 (Darban & Valipour, 2022) |
| speech recognition | Audio | LibriSepech$_{mini}$ (Panayotov et al., 2015) | Word Error Rate (↓) | 0.0134 (Zhang et al., 2020) |

Table 1: **Tasks included in DeepBench.** Datasets denoted by $_{mini}$ are the datasets reduced to 10,000 training set and 1,000 validation set due to massive size. PWC SOTA does not represent SOTA on such cases.

## 4.1 TASK DESCRIPTION

In DeepBench, the description of the task includes information on three components: the task to solve, the dataset to train and test the model on, and the metric to be used for evaluation. Since the tasks and metrics are basic and LLMs are expected to understand well, we only include a simple description of the task, e.g. "image classification task on CIFAR-10 dataset" and "The performance must be evaluated using accuracy". For loading the datasets, we provide two sources of the datasets: local storage and huggingface. In either case, we provide detailed information of the structure of the path and the dataset, as shown in the example of Figure 5. The types of tasks and corresponding metrics are specified on the Table 1.

## 4.2 EXECUTION ENVIRONMENTS

To minimize the effect of the debugger and to solely focus on the ability to generate high-performing architecture, we provide an experimental environment of a temporary Conda[4] virtual environment. Virtual environment for the execution of the program includes basic external packages like Tensorflow or Pytorch. By providing a compatible environment for most cases, we lower the possibility of falling into the pitfalls of environmental problem.

---

[4]https://conda.io/

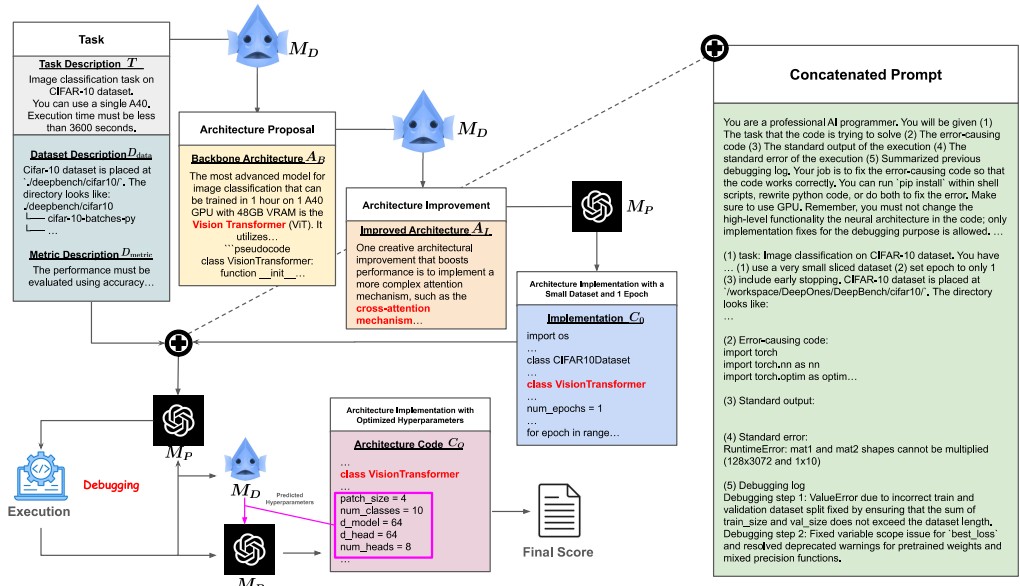

Figure 5: **Overview of DeepBench evaluation scheme for the task of image classification on CIFAR-10 dataset**: $M_A$ is an architect model which designs, proposes, or improve the deep neural architecture. On the other hand, $M_P$ is a programmer agent which has a lot of knowledge in AI and programming capability - here we uniformly use GPT-4o. Once the architecture generated by $M_A$ is successfully implemented and validated through small portion of data and 1 epoch, $M_A$ suggests a set of optimal hyperparameters for the model. Then the code is tested in a scaled-up scenario.

## 4.3 EVALUATION PIPELINE

In this section, we discuss each stage in our evaluation pipeline. Whole pipeline is visualized on Figure 5, while generated examples are available in the Appendix B.3.

### 4.3.1 TASK SPECIFICATION

To assess the model's ability to design programs, we generate a fully executable implementation of the proposed design, which we refer to as deep learning program synthesis. The task of deep learning program synthesis can be formally defined as:

$$P, S = M(T) \tag{1}$$

where the program $P$ and the metric score $S$ is generated from the natural language description of the task $T$.

In our evaluation pipeline, we break down this mapping into three key stages: requirement-based architecture proposal, architecture improvement, and evaluation. Each of these steps is explained below.

### 4.3.2 REQUIREMENT-BASED ARCHITECTURE PROPOSAL

In practical scenarios where we want to apply deep learning programs, the task is not the only consideration; we often face constraints related to GPU resources and time. The initial step involves proposing a base model that can address the task while adhering to the constraints of time and GPU availability. Thus, we provide the designer model $M_D$ with the task to be solved and the relevant constraints regarding GPU and time. Based on this information, $M_D$ determines the most appropriate backbone model architecture $A_B$ to use. This process is formalized as:

$$A_B = M_D(T) \quad \text{where} \quad T = \{\text{task description}, \text{requirements}\} \tag{2}$$

### 4.3.3 ARCHITECTURE IMPROVEMENT

Proposing an architecture is appropriate for evaluating the model's proposal capability, but not for program designing capability. We instruct $M_D$ to modify $A_B$ to enhance performance and efficiency, resulting in an improved architecture $A_I$. This approach mirrors the typical process used by most researchers, where they modify existing models to their own uses. This is expected to have a more significant impact than simple layer modifications or hyperparameter tuning performed in previous works. Additionally, our models generate reference Python code or Python-style pseudocode $C_r$ to facilitate the next step of implementation. This process is formalized as:

$$A_I, C_R = M_D(A_B) \tag{3}$$

### 4.3.4 ARCHITECTURE IMPLEMENTATION

Using the detailed description of the improved architecture, the programmer model $M_P$ tries to implement it by writing an executable Python code $C_0$. Since testing the validity of $C_0$ with the entire dataset can take a long time, we start with a small subset of the data for training and run just one epoch training. This lets us quickly check whether $C_0$ is written correctly and can run without issues. If $C_0$ is invalid, $M_P$ goes through a process of iterative debugging, where each step is labeled as $i$, to produce the $i$-th debugged implementation $C_i$. We provide rich information for debugging, including the original task description $T$, error-causing code $C_{i-1}$, standard output $O_{i-1}$, standard error $E_{i-1}$, and debugging log $L_{i-1}$. The debugging log is created and updated for every debugging step with a simple prompt `summarize the problem and your solution in one line natural language sentence, like Syntax Error -> changed the line ‘print("hello world"‘ to ‘print("hello world")‘`. This memory prevents $M_P$ repeating the same debugging which does not resolve a problem. Once a numeric score $S_i$ is successfully recorded in the log file, we consider $C_i$ is validated. This process can be expressed as:

$$C_0 = M_P(T, A_I) \quad \text{(Implementation)} \tag{4}$$
$$C_i, S_i, L_i = M_P(T, C_{i-1}, O_{i-1}, E_{i-1}, L_1, ..., L_{i-1}) \quad \text{(debugging)} \tag{5}$$

### 4.3.5 PROGRAM EVALUATION

After validation, $M_P$ updates the code to use the full dataset and the best hyperparameters based on the recommendations from $M_D$ regarding the set of optimal hyperparameters $H_O$. As a result, we obtain a correctly implemented architecture along with the training and testing code, namely $C_O$ that includes suitable hyperparameters. In addition, we emphasize the time requirements so that the code stops training and starts evaluation after the predefined time limitation is past.

$$H_O = M_D(T, A_I, C_i) \quad \text{(Hyperparameter Recommendation)} \tag{6}$$
$$C_O = M_P(C_i, H_O, T) \quad \text{(Complete Evaluation Code)} \tag{7}$$

## 5 EXPERIMENT

### 5.1 MODEL TRAINING

We use Llama3.1-8B as our base model. We train Llama3.1-8B on DeepData, with two-stage training. On the first stage, we train it on a general instruction data. On the second stage, given that the model has sufficiently learned AI-realated knowledge, we further train it on DLPD data, which includes requirement-based architecture proposing, property-based architecture improvement, and hyperparameter prediction. For instruction-tuning stage, we have used the batch size of 16, while for DLPD-tuning stage, we have used the batch size of 4 due to long data. In addition, for DLPD-tuning stage, we replayed randomly sampled 1% of instruction data to prevent catastrophic forgetting. For both training stages, we trained the models for 2 epochs, using 1 percent of the dataset for warmup steps, learning rate of 3e-4, cosine learning rate decay, and Adam-mini (Zhang et al., 2024) optimizer. This took around 3 days on 4 NVIDIA A6000 GPUs. Furthermore, to accelerate training and inference speeds and to reduce the memory usage of LLMs, we have applied several techniques. We employed LLaMA-Factory (Zheng et al., 2024), flash attention 2 (Dao, 2023), unsloth[5], and lora plus (Hayou et al., 2024) for acceleration.

---

[5] `https://github.com/unslothai/unsloth`

## 5.2 EXPERIMENTAL DETAILS

In our experiment, we compare DeepLlama-8B, Llama3.1-8B fine-tuned on DeepData, to Llama3.1-8B-Instruct, GPT-4o and Claude3.5-Sonnet. As an agent for implementation and debugging, we utilize GPT-4o uniformly. Following the setting of MLAgentBench (Huang et al., 2023), we iteratively run the experiment for 8 times to mitigate the randomness of LLMs. We use single NVIDIA A40 GPU with 48GB for each run, with 4200 seconds of a program execuion time limit and 3600 seconds of training time limit, and the limit of 20 debugging phases.

| Task | DeepLlama-8B | DeepLlama-8B+ | Llama3.1-8B-Instruct | Llama3.1-8B-Instruct+ | GPT-4o | GPT-4o+ | Claude 3.5 Sonnet | Claude 3.5 Sonnet+ |
|---|---|---|---|---|---|---|---|---|
| image classification (↑) | 68.66 | 64.33 | 61.73 | **81.93** | 71.02 | 10.00 | 49.47 | 70.23 |
| best/worst | **96.31**/37.50 | 85.84/45.64 | 94.21/10.59 | 89.68/74.35 | 95.75/9.33 | 10.00/10.00 | 84.73/13.69 | 71.16/69.08 |
| execution time of the best run (sec) | 4200.35 | 3634.73 | 1588.68 | 600.65 | 3686.25 | 4191.70 | 513.48 | 110.15 |
| success rate (%) | 62.50 | 87.50 | 75.00 | 62.50 | 100.00 | 12.50 | 37.50 | 37.50 |
| text-to-image generation (↓) | - | - | 397.29 | 533.33 | - | - | - | - |
| best/worst | - | - | 397.29/397.29 | 533.33/533.33 | - | - | - | - |
| execution time of the best run (sec) | - | - | 3318.77 | 1795.38 | - | - | - | - |
| success rate (%) | - | - | 12.50 | 12.50 | - | - | - | - |
| image captioning (↑) | - | - | 0.07 | - | - | **0.62** | - | 0.08 |
| best/worst | - | - | 0.07/0.07 | - | - | 0.62/0.62 | - | 0.08/0.08 |
| execution time of the best run (sec) | - | - | 1256.49 | - | - | 3722.14 | - | 4178.90 |
| success rate (%) | - | - | 12.50 | - | - | 12.50 | - | 12.50 |
| object detection (↑) | - | **55.77** | - | - | - | - | - | 0.06 |
| best/worst | - | 55.77/55.77 | - | - | - | - | - | 0.06/0.06 |
| execution time of the best run (sec) | - | 3765.00 | - | - | - | - | - | 3677.12 |
| success rate (%) | - | 12.50 | - | - | - | - | - | 12.50 |
| face recognition (↑) | 3.49 | 3.98 | 4.01 | 7.04 | 24.41 | 4.97 | 19.88 | 38.36 |
| best/worst | 5.78/1.77 | 4.46/3.51 | 4.01/4.01 | 10.05/4.04 | 74.53/4.19 | 4.97/4.97 | 39.67/0.08 | 89.25/5.78 |
| execution time of the best run (sec) | 51.61 | 848.49 | 2132.60 | 1005.46 | 842.44 | 3675.96 | 2799.42 | 2048.41 |
| success rate (%) | 62.50 | 25.00 | 12.50 | 25.00 | 62.50 | 12.50 | 25.00 | 62.50 |
| question answering (↑) | 45.62 | 90.87 | 79.72 | 69.35 | 88.47 | 88.92 | 87.05 | 66.09 |
| best/worst | 45.62/45.62 | 91.12/90.48 | 92.33/50.54 | 88.17/50.54 | 88.63/88.28 | 89.46/88.58 | 88.27/85.83 | 89.09/50.54 |
| execution time of the best run (sec) | 150.22 | 3626.00 | 3741.63 | 1173.90 | 875.21 | 2525.95 | 1431.99 | 3648.07 |
| success rate (%) | 12.50 | 50.00 | 50.00 | 25.00 | 37.50 | 37.50 | 25.00 | 50.00 |
| sentiment classification (↑) | 81.54 | 86.80 | 91.29 | 90.33 | 90.14 | 64.18 | 72.22 | 84.24 |
| best/worst | 93.69/50.80 | 94.27/50.92 | 94.50/89.68 | 95.07/83.60 | 91.97/87.96 | 90.48/50.92 | 95.30/50.92 | 94.38/63.42 |
| execution time of the best run (sec) | 1286.53 | 863.23 | 2064.33 | 4200.30 | 1045.23 | 1387.49 | 3629.44 | 2937.68 |
| success rate (%) | 50.00 | 87.50 | 37.50 | 37.50 | 75.00 | 37.50 | 50.00 | 62.50 |
| natural language inference (↑) | 83.00 | 67.40 | 70.92 | 31.82 | 78.70 | 79.35 | 65.63 | 80.66 |
| best/worst | 84.20/81.79 | 87.77/32.59 | 87.06/35.41 | 31.82/31.82 | 82.50/74.81 | 80.23/78.47 | 82.52/40.79 | 89.19/64.68 |
| execution time of the best run (sec) | 2251.26 | 107.68 | 3658.84 | 4200.53 | 3221.65 | 3555.39 | 3703.65 | 4028.57 |
| success rate (%) | 25.00 | 75.00 | 50.00 | 12.50 | 50.00 | 25.00 | 37.50 | 50.00 |
| recommendation system (↓) | 1.10 | 1.02 | 1.90 | 1.07 | 1.04 | 437.11 | 0.97 | 1.11 |
| best/worst | 0.96/1.41 | 0.97/1.07 | 0.98/2.86 | 1.07/1.07 | 0.95/1.28 | 0.95/1309.36 | 0.97/0.97 | 1.11/1.11 |
| execution time of the best run (sec) | 925.31 | 1267.56 | 52.89 | 73.40 | 12.20 | 537.82 | 120.45 | 259.78 |
| success rate (%) | 75.00 | 62.50 | 50.00 | 12.50 | 50.00 | 37.50 | 12.50 | 12.50 |
| speech recognition (↓) | - | - | - | - | - | - | - | - |
| best/worst | - | - | - | - | - | - | - | - |
| execution time of the best run (sec) | - | - | - | - | - | - | - | - |
| success rate (%) | - | - | - | - | - | - | - | - |

Table 2: **Comparison of the baselines.** A + sign indicates architectural improvement has been additionally performed. The first row of each task shows the mean value of each assigned metric. A **bold** score represents the best result, while an underlined score represents the second best. Success rates are calculated over 8 iterations. Scores highlighted in red indicate a performance decrease due to architectural improvements, while scores highlighted in blue indicate a performance increase resulting from these improvements. Darker color indicates greater performance increase or decrease.

## 5.3 RESULTS AND ANALYSIS

### 5.3.1 QUANTITATIVE ANALYSIS

Table 2 displays the performance of the designed architecture across 8 different scenarios. The models DeepLlama-8B, Llama3.1-8B-Instruct, GPT-4o, and Claude-3.5-Sonnet are first evaluated based on their proposed architectures, which are mostly existing ones. In this initial trial, DeepLlama-8B, GPT-4o, Claude-3.5-Sonnet show comparable performance in both mean metrics and best metrics, while Llama-8B-Instruct is far behind. To evaluate whether the models truly understand the architectures, we instruct them to enhance both performance and efficiency through improvements. Llama3.1-8B-Instruct and GPT-4o mostly fails to improve the architectures in terms of mean score, best score, execution time, and success rate, implying it is not proposing valid improvements to the architecture. In contrast, DeepLlama-8B and Claude-3.5-Sonnet succeed in improving both metric scores and efficiency in many cases.

### 5.3.2 QUALITATIVE ANALYSIS

**Quality of Architecture Design** The choice of backbone model significantly impacts performance in many cases. For instance, a ViT model without modifications can outperform a heavily modified ResNet. Here, we examine the backbone models proposed by each architectural model. Table 3 shows that DeepLlama introduces cutting-edge technologies such as Mamba Blocks, LLaVA, and RAVEN. Since these models are known to operate under the constraints of the experiment with a single A40 and 48 VRAM, suggestions are quite reasonable.

| Task | Category | DeepLlama-8B | Llama3.1-8B-Instruct | GPT-4o | Claude 3.5 Sonnet |
|---|---|---|---|---|---|
| image classification | most common model | Vision Transformer | EfficientNet | EfficientNet | EfficientNet |
| | most common improvement | Mamba Blocks | Self-Attention | Mixed Precision | Squeeze-and-Excitation Blocks |
| | best case | Vision Transformer | EfficientNet | EfficientNet-B1 | EfficientNet-B0 |
| | | - | - | - | - |
| text-to-image-generation | most common model | Stable Diffusion | DALL-E 2 | Stable Diffusion | ControlNet |
| | most common improvement | Textual Inversion | Hierarchical Transformer Encoder | Mixed Precision | Attention Mechanism |
| | best case | - | Stable Diffusion | Stable Diffusion | - |
| | | - | - | BERT text encoder | - |
| image captioning | most common model | LLaVA-1.5 | Vision Transformer | CLIP | BLIP |
| | most common improvement | RAVEN | Knowledge Distillation | Mixed Precision | Self-Attention |
| | best case | - | - | Transformer | ViT + GPT-2 |
| | | - | - | EfficientNet Feature Extractor | Additional Attention |
| object detection | most common model | YOLOv8 | YOLOv8 | YOLOv8 | YOLOv5 |
| | most common improvement | Attention Mechanism | Multi-Scale Feature Fusion | Mixed Precision | Feature Pyramid Network |
| | best case | YOLOv8 | - | - | - |
| | | Multi-scale Testing, MobileNet V3 Backbone | - | - | - |
| face recognition | most common model | FaceNet | ArcFace | ArcFace | ArcFace |
| | most common improvement | Group Convolution | Knowledge Distillation | Mixed Precision | Attention Mechanism |
| | best case | ResNet | ResNet | ResNet-50 | ArcFace |
| | | Hierarchical Constrastive Function | Knowledge Distillation | - | MobileNet-V3 Feature Extractor |
| question answering | most common model | ChatGPT | DistilBERT | DistilBERT | DistilBERT |
| | most common improvement | Pruning | Knowledge Distillation | Mixed Precision | Ensemble Learning |
| | best case | BERT | DistilBERT | DistilBERT | DistilBERT |
| | | Auxiliary Task Learning | DistilBERT | Knowledge Distillation | Ensemble Knowledge Distillation |
| sentiment classification | most common model | GPT2 | DistilBERT | DistilBERT | DistilBERT |
| | most common improvement | Lightweight Attention | Knowledge Distillation | Mixed Precision | Progressive Layer Dropping |
| | best case | DeBERTa-V3 | DistilBERT | BERT | DistilBERT |
| | | Layer Normalization | Knowledge Distillation | - | - |
| natural language inference | most common model | LLaMA-2-7B | BERT | DistilBERT | DistilBERT |
| | most common improvement | Multi-task Learning | Multi-Task Learning | Mixed Precision | Progressive Layer Dropping |
| | best case | RoBERTa | DistilBERT | DistilBERT | DeBERTa-V3 |
| | | Chain-of-Thoughts, Few-shot | DistilBERT | DistilBERT | Attention Fusion, Progressive Unfreezing |
| recommendation system | most common model | Llama-2-7B | Neural Collaborative Filtering | LightFM | LightGCN |
| | most common improvement | QLora | Attention Mechanism | Mixed Precision | Attention Mechanism |
| | best case | KATRec | Neural Collaborative Filtering | Neural Collaborative Filtering | Neural Collaborative Filtering |
| | | Integration with BERT | - | - | - |
| speech recognition | most common model | Whisper | Wav2Vec | Wav2Vec | Wav2Vec |
| | most common improvement | Attention Mechanism | Attention Mechanism | Mixed Precision | Attention Mechanism |
| | best case | - | - | - | - |
| | | - | - | - | - |

Table 3: **Models and improvements suggested by the designer models.** *Most common models* and *most common improvements* are investigated over 8 runs. If there are multiple most-common ones found, we denoted the one with better performance.

On the other hand, GPT-4o exhibits a bias towards mixed precision, while Llama3.1-8B-Instruct proposes knowledge distillation as an improvement for most of the tasks. In contrast, DeepLlama and Claude-3.5-Sonnet generate a wider variety of improvements, effectively improving the model performance at the same time.

**Implementation's Correspondence to the Proposed Program Design**   Although the quality of designed architecture is reasonable, the Table 3 shows that the best case occurs mostly using classical models, like BERT variants. This is mostly due to the discrepancy of implementation capabilities of the programmer model, GPT-4o. As shown in the Figure 4 suggests, GPT-4o already has substantial knowledge in AI-related technologies. Nevertheless, we observed that GPT-4o lacks knowledge on implementations of several recent techniques. For example, GPT-4o fails to apply common suggestions from DeepLlama, such as Mamba, LLaVA, or Llama. This affects DeepLlama negatively, as its initial suggestions are more relevant to cutting-edge technologies. Thus, while DeepBench effectively evaluates the model's ability to understand the architecture and generate appropriate improvements, the evaluation on architectural proposals appears limited.

# 6 CONCLUSION AND LIMITATIONS

In this paper, we introduce a comprehensive solution for the task of deep learning program design, by proposing a novel dataset and two benchmarks. Through the evaluation on DeepBench, we showcase that DeepData is effective for training a LLM to obtain a broad knowledge on architectures and techniques. Along with a quantitative and qualitative analysis and open-sourcing, we believe that this contributes to the more active research on the task of deep learning program design.

**Limitations**   Even we have shown DeepData and DeepBench's strength, there remains some limitations to be resolved. First, as mentioned in the qualitative study, GPT-4o often fails in implementing recent knowledge, even though it possesses one of the best code generation capabilities. This may lead to a distorted evaluation of the model's capability on architecture proposal. In addition, we rely on a closed-source models for code generation, which is extremely costly. In future works, we would like to suggest lightweight open-source models that can replace GPT-4o in deep learning programming task. These limitations pose the need for more investigations on the task of deep learning program design.

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

# A CATEGORY DISTRIBUTION

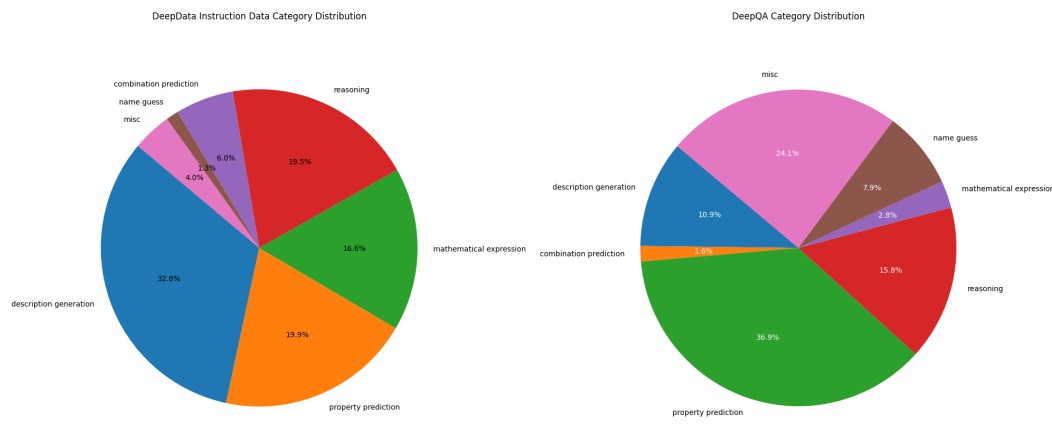

(a) **Category distribution of DeepData.**   (b) **Category distribution of DeepQA.**

Figure 6: Category distributions of DeepData and DeepQA.

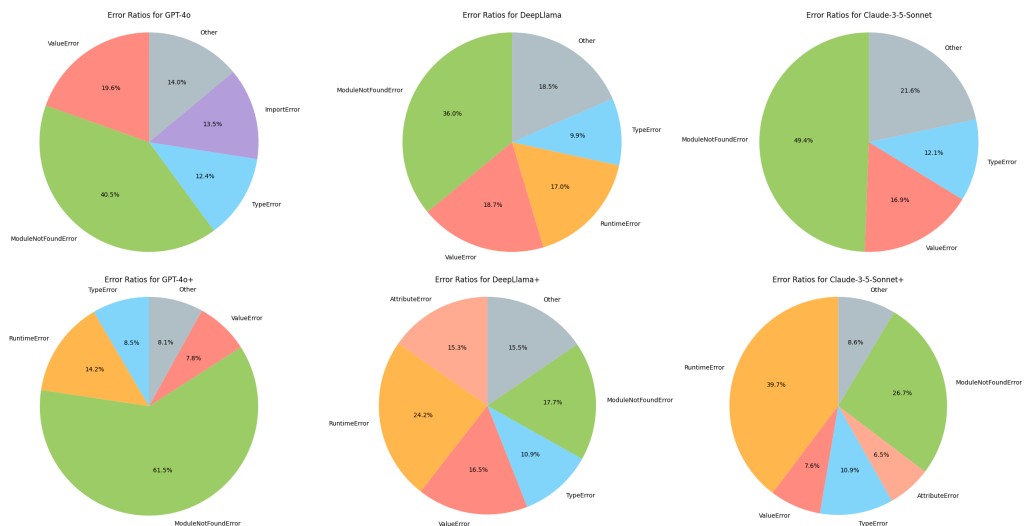

Figure 7: **Error ratio of the models.** GPT-4o primarily encounters ModuleNotFoundError, indicating a significant environmental configuration issue. Notably, GPT-4o displays a contrasting error pattern compared to DeepLlama and Claude-3.5-Sonnet, as the occurrence of ModuleNotFoundError increases substantially when it attempts to enhance the architecture.

## B EXPERIMENTAL DETAILS

### B.1 ERROR ANALYSIS

Figure 7 shows the percentage of errors encountered during the debugging process. A key observation is that GPT-4o encounters more instances of ModuleNotFoundError when trying to improve the architecture. This suggests that it relies heavily on external packages or libraries for its enhancements. In comparison, DeepLlama and Claude-3.5-Sonnet generate more RuntimeError, AttributeError, ValueError, and TypeError, which are mainly due to implementation issues. This indicates that while DeepLlama and Claude-3.5-Sonnet propose valid architectural improvements, GPT-4o primarily tries to use existing methods.

### B.2 FULL EXAMPLES

#### B.2.1 DEEPDATA

**Description Generation**

```
Could you explain the Glimpse-Based Actor-Critic (GBAC) model?
=> GBAC is a reinforcement learning model that integrates a hard
   attention mechanism into its architecture to process visual
   input more efficiently. Unlike conventional RL models that
   analyze the entire image, GBAC focuses on 'glimpses'-specific,
    high-resolution sections of the input frame. This allows GBAC
    to achieve significant computational advantages while
   maintaining competitive performance on tasks like video game
   environments. The architecture consists of three main networks
   : the Glimpse Network, which extracts features from the
   selected glimpse, the Action Network, which determines the
   action to execute, and the Location Network, which selects the
    next coordinates to observe.
```

**Combination Prediction**

```
How can we combine the Dynamic Grained Encoder with other
    transformer frameworks?
=> To combine the Dynamic Grained Encoder (DGE) with other
    transformer frameworks, one would typically integrate the DGE
    block into the existing architecture, replacing traditional
    encoder blocks with the adaptive mechanism in DGE. This not
    only hones the query generation based on region-specific
    features but also allows for performance optimization across
    tasks such as image classification or object detection. The
    synergy comes from DGE's ability to adjust its processing
    based on real-time input, allowing existing frameworks to
    become more efficient without necessitating a complete
    overhaul of their structures.
```

**Property Prediction**

```
What benefits does the hard attention mechanism provide in GBAC?
=> The hard attention mechanism in GBAC allows the model to
    stochastically select which parts of the frame to process,
    focusing computational resources on the most important visual
    stimuli. This results in improved efficiency since the model
    does not waste resources analyzing irrelevant details outside
    the selected glimpses. This strategy not only simulates human-
    like selective attention but also enables the model to operate
     effectively with less input data, thus speeding up training
    and reducing the overall computational burden required.
```

**Reasoning**

```
Explain how the GBAC model can match the performance of PPO
    despite processing fewer pixels.
=> GBAC matches the performance of the PPO model by intelligently
    selecting and focusing on relevant glimpses of the input frame
    , leading to enhanced training efficiency without sacrificing
    performance. By filtering out less relevant information, GBAC
    not only reduces the noise present in the input but also
    improves learning signals by allowing the model to concentrate
     on critical visual details essential for decision-making.
    This strategic processing allows GBAC to achieve competitive
    results even while handling a substantially reduced pixel
    count.
```

**Mathematical Expression**

```
Describe the Proximal Policy Optimization (PPO) algorithm
    mathematically.
=> The objective of the PPO algorithm is represented as follows: L
    ^{CLIP}(\theta) = \\hat{\\mathbb{E}}_{t} \\Big[ \\min(r_{t}(\
    theta)\\hat{A}_{t}, \\mathrm{clip}(r_{t}(\theta), 1 - \\
    epsilon, 1 + \\epsilon)\\hat{A}_{t}) \\Big]. Here, r_{t}(\
    theta) is the probability ratio of the action probabilities
    under the new policy to those under the old policy. \\hat{A}_{
    t} is the advantage function estimator. The clipping mechanism
     helps to prevent large updates in the policy, allowing for
    stable training by keeping the updates small and controlled.
```

**Name Guess**

```
Is there a technique that combines hard attention and
    reinforcement learning with efficient computation?
=> Yes, the Glimpse-Based Actor-Critic (GBAC) model exemplifies a
    technique that combines a hard attention mechanism with model-
    free reinforcement learning, specifically using the PPO
    algorithm. The key aspects of GBAC allow for efficient visual
    processing through selective attention while maintaining the
    core reinforcement learning principles of action selection and
     learning from the environment. This approach ensures that the
     performance does not heavily rely on the size of input images
     but rather on the strategically chosen glimpses that carry
    relevant information.
```

**Miscellaneous**

```
What are potential applications of measuring game distance in
    Ludii?
=> Applications of measuring game distance in Ludii include
    improving the performance of game-playing agents by leveraging
     established knowledge from similar games, recommending new
    games to users based on their preferences, and conducting
    transfer learning across games with similar dynamics. These
    applications facilitate a deeper understanding of the Ludii
    framework and enhance user engagement by suggesting games
    aligned with players' interests.
```

### B.2.2 DEEPQA

**Description Generation**

```
What is the main function of TASD in natural language processing?
A. To improve data compression techniques.
B. To generate natural language descriptions from tables.
C. To translate languages.
D. To summarize long texts.
E. To develop new table structures
=> B. To generate natural language descriptions from tables.
```

**Combination Prediction**

```
Which of the following is a method to reduce localization errors
    in AUVs?
A. Utilizing only visual information.
B. Implementing high-cost INS systems.
C. Using cooperative localization strategies.
D. Relying solely on dead reckoning.
E. Employing long-range sonar without additional sensors
=> C. Using cooperative localization strategies.
```

**Property Prediction**

```
What advantage does relaxed-LSS provide over traditional leverage
    score sampling?
A. It incorporates low-leverage data points for improved
    robustness.
B. It is computationally more intensive than traditional methods.
C. It only selects high-leverage points.
D. It samples data independently of their characteristics.
E. It offers no significant change from leverage score sampling
```

```
=> A. It incorporates low-leverage data points for improved
   robustness.
```

**Reasoning**

```
How does focal loss improve model performance in classification
    tasks?
A. It reduces training time significantly.
B. It increases the weight of well-classified examples.
C. It focuses on hard-to-classify cases in imbalanced datasets.
D. It simplifies the model architecture.
E. It applies only to binary classification tasks
=> C. It focuses on hard-to-classify cases in imbalanced datasets.
```

**Mathematical Expression**

```
What theoretical dimension bound is necessary for reliably
    estimating set intersections in VSAs?
A. m = O(n log(d))
B. m = O(n)
C. m = O(n log(M/\epsilon))
D. m = O(\sqrt{n})
E. m = O(d^2)
=> C. m = O(n log(M/\epsilon))
```

**Name Guess**

```
Which method does FewSOME utilize to prevent representational
    collapse?
A. Augmenting training data with noise.
B. Introducing a batch normalization technique.
C. A Loss Function named Stop Loss (L_{stop}).
D. Increasing the architecture depth.
E. Applying dropout layers
=> C. A Loss Function named Stop Loss (L_{stop}).
```

**Miscellaneous**

```
What is the result of using the proposed relational network on the
    visual XOR task?
A. The network is unable to learn the task.
B. The network achieves high error rates.
C. The network successfully generalizes relationships.
D. The network performs poorly compared to STDP approaches.
E. The network requires extensive parameter tuning
=> C. The network successfully generalizes relationships.
```

## B.3 EXAMPLE RUN OF DEEPLLAMA-8B

### B.3.1 REQUIREMENT-BASED ARCHITECTURE PROPOSAL

```
One of the most advanced models for natural language inference is
    InstructGPT. It InstructGPT is an instruction-tuned large
    language model that can perform various nlp tasks, including
    natural language inference, and it has been shown to achieve
    superior performance on this task compared to other models
    like GPT-3.5 and GPT-4.
Here is a Python code snippet for natural language inference using
    InstructGPT:

```pseudocode
class InstructGPT:
    function __init__(tokenizer, model_name_or_path, max_len=512,
        max_query_len=64, max_context_len=512, temperature=1.0,
        top_k=100, top_p=0.1, top_n=10, num_queries=20,
        num_contexts=5, num_answers=5, device="cuda", logger=None)
        :
         # Initialize model parameters
         set device to the specified device
         set logger to the specified logger
         initialize tokenizer with the provided tokenizer
         set max_len to the provided maximum length
         set max_query_len to the provided maximum query length
         set max_context_len to the provided maximum context length
         set temperature to the provided temperature
         set top_k to the provided top k value
         set top_p to the provided top p value
         set top_n to the provided top n value
         set num_queries to the provided number of queries
         set num_contexts to the provided number of contexts
         set num_answers to the provided number of answers

         # Load the model from the specified path
         load model using the specified model_name_or_path
         move model to the specified device
         set model to evaluation mode
         set tokenizer to evaluation mode

         # Load a sample context from a specified path
         load sample context from the provided path

         # Initialize contexts and queries as empty lists
         initialize contexts as empty list
         initialize queries as empty list

         # Create a template for the context and queries
         create context template with num_contexts and num_answers
         create query template with num_queries

         # Initialize answer template
         create answer template

         # Prepare context and query lists
         generate context examples and append them to contexts
         generate query examples and append them to queries
```

```
1080        # Save generated contexts and queries to files
1081        save contexts to the specified file
1082        save queries to the specified file
1083
1084        # Prepare the prompt for the model
1085        create prompt with context template, query template, and
1086            answer template
1087        set prompt to the model's prefix
1088
1089        # Log the number of contexts and queries generated
1090        log the count of generated contexts
1091        log the count of generated queries
1092
1093    function __call__(context, query):
1094        # Convert context and query to tokenized format
1095        convert context to tokenized format
1096        convert query to tokenized format
1097
1098        # Combine context and query to form the complete prompt
1099        create complete prompt by concatenating context and query
1100
1101        # Evaluate the model with the prompt and get the output
1102        output = evaluate model using complete prompt and
1103            temperature
1104
1105        # Tokenize output and return it
1106        return tokenized output
```

### B.3.2 ARCHITECTURE IMPROVEMENT

```
One creative architectural improvement to boost performance is the
    use of Chain-of-Thought (CoT) prompting. This method involves
    prompting the model to provide step-by-step reasoning before
    arriving at a conclusion, which helps improve accuracy.
    Additionally, using a chain-of-thought prompt can be combined
    with other techniques like few-shot learning to further
    enhance performance.
class InstructGPT:
    """Class to interact with InstructGPT API."""

    function __init__(prompt, num_prompt_retries=1, **kwargs):
        """Initialize the InstructGPT model with the given prompt
            and optional retries."""
        self._prompt = prompt
        self._num_prompt_retries = num_prompt_retries
        self._kwargs = kwargs

    function _call_api(api_key, prompt, num_prompt_retries):
        """Call the InstructGPT API with the provided prompt and
            retry settings."""
        if num_prompt_retries is None:
            num_prompt_retries = self._num_prompt_retries

        try:
            response = call InstructGPT API with the provided API
                key, prompt, and retry settings
            return response
```

```
except Exception as e:
    log error and retry num_prompt_retries times, then
        raise the exception

function call_api(prompt):
    """Call the InstructGPT API with the given prompt."""
    return _call_api(self.api_key, self._prompt, self.
        _num_prompt_retries)

function call_few_shot(prompt, few_shot_prompts):
    """Call the InstructGPT API with a few-shot prompt and
        provided examples."""
    if few_shot_prompts is None:
        few_shot_prompts = list of few-shot prompts

    return _call_api(self.api_key, prompt with few-shot
        examples, self._num_prompt_retries)

function call_chain_of_thought(prompt, chain_of_thought_prompt
    ):
    """Call the InstructGPT API with a chain-of-thought prompt
        ."""
    return _call_api(self.api_key, prompt with chain-of-
        thought reasoning, self._num_prompt_retries)
```
```

### B.3.3 ARCHITECTURE IMPLEMENTATION

```
import torch
from transformers import pipeline,
    AutoModelForSequenceClassification, AutoTokenizer
from datasets import load_dataset
import os
from sklearn.metrics import accuracy_score

# Set up device
device = torch.device("cuda" if torch.cuda.is_available() else "
    cpu")

# Load dataset
dataset = load_dataset('glue', 'mnli')

# Function to get a small slice of the dataset for quick testing
def get_small_dataset_slice(dataset, split='train', size=100):
    return dataset[split].select(range(size))

train_dataset = get_small_dataset_slice(dataset, 'train')
validation_dataset = get_small_dataset_slice(dataset, '
    validation_matched')

# Load pre-trained model and tokenizer
model_name = "roberta-large-mnli"
tokenizer = AutoTokenizer.from_pretrained(model_name)
model = AutoModelForSequenceClassification.from_pretrained(
    model_name).to(device)

# Preparing the inference pipeline with the model on GPU
```

```
1188  nlp = pipeline("text-classification", model=model, tokenizer=
1189      tokenizer, device=0)
1190
1191  # Function for Chain-of-Thought (CoT) prompting
1192  def chain_of_thought_prompts(premise, hypothesis):
1193      return f"Given the premise and hypothesis below, provide the
1194          reasoning chain leading to the conclusion:\nPremise: {
1195          premise}\nHypothesis: {hypothesis}\nReasoning:"
1196
1197  # Few-shot learning examples
1198  few_shot_examples = [
1199      "Premise: A soccer game with multiple males playing.\
1200          nHypothesis: Some men are playing a sport.\nLabel:
1201          entailment\n",
1202      "Premise: An older and younger man smiling.\nHypothesis: Two
1203          men are smiling and laughing at the cats playing on the
1204          floor.\nLabel: neutral\n",
1205      "Premise: A man inspects the uniform of a figure in some East
1206          Asian country.\nHypothesis: The man is sleeping.\nLabel:
1207          contradiction\n"
1208  ]
1209
1210  # Early stopping criteria
1211  early_stopping_patience = 1  # Minimal as we aim for quick test
1212  epochs = 1  # Only one epoch for quick testing
1213
1214  # Training loop mockup (assuming the InstructGPT API works
1215      similarly for demonstration)
1216  best_accuracy = 0
1217  patience_counter = 0
1218
1219  for epoch in range(epochs):
1220      preds = []
1221      labels = []
1222
1223      # Validation step
1224      for i, data in enumerate(validation_dataset):
1225          premise = data['premise']
1226          hypothesis = data['hypothesis']
1227          label = data['label']
1228
1229          cot_prompt = chain_of_thought_prompts(premise, hypothesis)
1230          cot_reasoning = "Reasoning: Considering the premise and
1231              hypothesis logically...\n"  # Simplified CoT
1232
1233          prompt = cot_prompt + cot_reasoning
1234
1235          few_shot_prompt = "".join(few_shot_examples) + prompt
1236
1237          # Using model to get the output (mockup InstructGPT call)
1238          result = nlp(premise + " " + hypothesis)
1239          pred_label = result[0]['label']
1240
1241          # Convert label to expected format (0: entailment, 1:
1242              neutral, 2: contradiction)
1243          if pred_label == "ENTAILMENT":
1244              pred = 0
1245          elif pred_label == "NEUTRAL":
1246              pred = 1
```

```
            else:
                pred = 2

            preds.append(pred)
            labels.append(label)

        current_accuracy = accuracy_score(labels, preds)
        if current_accuracy > best_accuracy:
            best_accuracy = current_accuracy
            patience_counter = 0
        else:
            patience_counter += 1

        if patience_counter >= early_stopping_patience:
            break

# Log the final accuracy
with open('./metric.log', 'w') as f:
    f.write(f"{best_accuracy:.4f}")

print(f"Final logged accuracy: {best_accuracy:.4f}")
```

### B.3.4 Generating Full Code with Optimal Hyperparameters

```
import torch
from transformers import pipeline,
    AutoModelForSequenceClassification, AutoTokenizer
from datasets import load_dataset
import os
from sklearn.metrics import accuracy_score
import time

# Set up device
device = torch.device("cuda" if torch.cuda.is_available() else "
    cpu")

# Load dataset
dataset = load_dataset('glue', 'mnli')
train_dataset = dataset['train']
validation_dataset = dataset['validation_matched']

# Load pre-trained model and tokenizer
model_name = "roberta-large-mnli"
tokenizer = AutoTokenizer.from_pretrained(model_name)
model = AutoModelForSequenceClassification.from_pretrained(
    model_name).to(device)

# Preparing the inference pipeline with the model on GPU
nlp = pipeline("text-classification", model=model, tokenizer=
    tokenizer, device=0)

# Function for Chain-of-Thought (CoT) prompting
def chain_of_thought_prompts(premise, hypothesis):
    return f"Given the premise and hypothesis below, provide the
        reasoning chain leading to the conclusion:\nPremise: {
        premise}\nHypothesis: {hypothesis}\nReasoning:"

# Few-shot learning examples
```

```
few_shot_examples = [
    "Premise: A soccer game with multiple males playing.\
        nHypothesis: Some men are playing a sport.\nLabel:
        entailment\n",
    "Premise: An older and younger man smiling.\nHypothesis: Two
        men are smiling and laughing at the cats playing on the
        floor.\nLabel: neutral\n",
    "Premise: A man inspects the uniform of a figure in some East
        Asian country.\nHypothesis: The man is sleeping.\nLabel:
        contradiction\n"
]

# Early stopping criteria
learning_rate = 0.0005  # Effective for fine-tuning transformer
    models
early_stopping_patience = 1  # Stop after one epoch without
    improvement
epochs = 1  # One epoch for quick testing

# Training settings
best_accuracy = 0
patience_counter = 0

start_time = time.time()
for epoch in range(epochs):
    preds = []
    labels = []

    # Validation step
    for i, data in enumerate(validation_dataset):
        premise = data['premise']
        hypothesis = data['hypothesis']
        label = data['label']

        cot_prompt = chain_of_thought_prompts(premise, hypothesis)
        cot_reasoning = "Reasoning: Considering the premise and
            hypothesis logically...\n"  # Simplified CoT

        prompt = cot_prompt + cot_reasoning

        few_shot_prompt = "".join(few_shot_examples) + prompt

        # Using model to get the output (mockup InstructGPT call)
        result = nlp(premise + " " + hypothesis)
        pred_label = result[0]['label']

        # Convert label to expected format (0: entailment, 1:
            neutral, 2: contradiction)
        if pred_label == "ENTAILMENT":
            pred = 0
        elif pred_label == "NEUTRAL":
            pred = 1
        else:
            pred = 2

        preds.append(pred)
        labels.append(label)

    current_accuracy = accuracy_score(labels, preds)
```

```
    if current_accuracy > best_accuracy:
        best_accuracy = current_accuracy
        patience_counter = 0
    else:
        patience_counter += 1

    if patience_counter >= early_stopping_patience:
        break

    # Check if training time exceeds 3600 seconds
    current_time = time.time()
    if current_time - start_time > 3600:
        print("Training exceeded 3600 seconds. Stopping early.")
        break

# Log the final accuracy
with open('./metric.log', 'w') as f:
    f.write(f"{best_accuracy:.4f}")

print(f"Final logged accuracy: {best_accuracy:.4f}")
```

