# OpenReview forum: "Designing Deep Learning Programs with Large Language Models"
_ICLR.cc/2025/Conference — ICLR 2025 Conference Withdrawn Submission_

### Official Review · Reviewer_LyMe · 2024-10-24

**Soundness:** 3
**Presentation:** 3
**Contribution:** 2
**Rating:** 3
**Confidence:** 4

**Summary:**

This paper studies Deep Learning Program Design (DLPD), in which language models design programs that utilize deep learning to solve specific tasks. The authors developed a training dataset, DeepData, as well as two benchmarks, DeepQA and DeepBench. The authors evaluated several language models, including one that was fine-tuned on their dataset.

**Strengths:**

[S1] This paper highlights an interesting problem, DLPD.

[S2] This paper makes a valuable contribution to the community by releasing a new dataset and two new benchmarks for DLPD, which are expected to accelerate further research in this field.

**Weaknesses:**

[W1] A discrepancy exists between the issue highlighted at the beginning and the solution the authors proposed. The authors state, “Current methods for automating the synthesis of deep learning programs often rely on basic code templates.” Although this is generally accurate, the new DeepBench does not significantly differ from these previous approaches. The tasks are standard close-ended ones (such as image classification and face recognition). Furthermore, the evaluated language models merely generate the neural network architectures and hyperparameters in a fixed way, and all other aspects of programming are always delegated to GPT-4o (Figure 5). This is essentially quite similar to merely using templates. For example, would it be possible to make the benchmark more comprehensive and general to accommodate a wider range of approaches, such as new loss functions, ensembling, stacking, etc?

[W2] DeepQA is a compelling new benchmark for those who are interested in deep learning programming. However, from the perspective of language model developers, this dataset holds limited appeal, as existing models like GPT-4o have already achieved a high-performance score of over 80%. This benchmark addresses a largely “solved” problem and fails to introduce any new technical challenges to the research community.

[W3] I also question whether the authors should define DeepBench as a benchmark for language models rather than a benchmark for programming agent systems. The authors’ pipeline (Figure 5) represents only one agent architecture for generating deep learning programs. It is commonly observed that such agent pipeline designs significantly influence the outcome. The evaluation results of LLMs depend on this architecture, but it remains unclear whether this design is optimal. Would it be possible to provide any rationale behind this design, or is it reasonable to generalize this benchmark as a programming agent benchmark?

[W4] The experimental results present few new insights. The results are generally unsurprising, with most of the findings aligning with prior expectations. For example, everyone already knows that Claude 3.5 Sonnet is a strong coder. This significantly limits the scientific contribution of the paper to the research community. For instance, it would be great if the authors could provide any insight specific to this task.

**Questions:**

[Q1] What are the noteworthy implications of your experimental results? I find that most of the results were somewhat expected. Do you believe there are any surprising findings and their potential implications for researchers or practitioners?

[Q2] Is it possible to break down the performance differences between models? For instance, while we observe an improvement in DeepLlama-8B over Llama-3.1-8B, what factors contributed to this enhancement, and to what extent did each factor influence the overall performance?

---

### Official Review · Reviewer_xL3a · 2024-10-29

**Soundness:** 3
**Presentation:** 3
**Contribution:** 3
**Rating:** 3
**Confidence:** 4

**Summary:**

This paper introduces the task of Deep Learning Program Design (DLPD), a task of designing effective deep learning programs using appropriate architectures and techniques tailored for specific tasks. It proposes Deep Ones, a comprehensive solution for DLPD, which includes a large-scale dataset, a lightweight benchmark, a model DeepLlama that is fine-tuned on the proposed dataset. On the benchmark, DeepLlama demonstrates better architecture suggestion capability than GPT-4o and better performance than Claude-3.5-Sonnet, demonstrating the effectiveness of Deep Ones.

**Strengths:**

This paper introduces the task of Deep Learning Program Design (DLPD), which is crucial due to AI's growing popularity. To bridge the gap between traditional code generation and deep learning architecture usage, it proposes a novel solution, Deep Ones, to design effective deep learning programs for specific tasks utilizing appropriate architectures and techniques. The authors propose a large-scale dataset, a multi-choice QA benchmark, and a lightweight benchmark specifically tailored for evaluating the program design capabilities. And the model DeepLlama fine-tuned on the proposed dataset shows effectiveness for DLPD, demonstrating better architecture suggestion capability than GPT-4o and better performance than Claude-3.5-Sonnet.

**Weaknesses:**

The main problem for this work is that the proposed model, DeepLlama, is not that effective and often underperforms compared to baseline models. For instance, as shown in Table 2, DeepLlama solves fewer tasks than its base model, Llama3.1-8B-Instruct, within the DeepBench benchmark. Of the six tasks it does solve, it underperforms in three cases, whether considering architecture improvements or not. This suggests that the fine-tuning process offers limited benefits for enhancing the base model's DLPD capabilities, raising concerns about the quality of the proposed training dataset and methodology. It would be helpful if the authors could provide more insight into why DeepLlama underperforms on certain tasks. Additionally, the paper contains several typos and errors. For instance, in Figure 5, the caption incorrectly labels $M_{A}$ as the architect model, while the figure shows $M_{D}$. And some figures, such as Figure 4, have low resolution. It would be better if the authors could provide higher-quality images in the final version.

**Questions:**

1. When fine-tuning using DLPD-style data, how is pseudocode incorporated into the DLPD data points? Could you provide more details?
2. As mentioned in the weaknesses, DeepLlama often underperforms compared to its base model. Have you conducted any error analysis to understand the reasons for the underperformance? Or have you considered alternative fine-tuning approaches that might improve results across more tasks?
3. You mention that one limitation of the proposed solution is the implementation capabilities of the programmer model, GPT-4o. Why don’t you try the OpenAI’s o1-preview model, which is the state-of-the-art model specifically designed for complex reasoning tasks including programming? It would be beneficial to discuss the potential benefits and challenges of using the o1-preview model compared to GPT-4o.

---

### Official Review · Reviewer_67Pd · 2024-11-01

**Soundness:** 2
**Presentation:** 3
**Contribution:** 3
**Rating:** 5
**Confidence:** 3

**Summary:**

The authors propose the Deep Ones, a solution for DLPD, comprised of three components: DeepData, DeepQA, and DeepBench.

For DeepData, the authors build a dataset from arXiv papers and corresponding implementations on GitHub, with a portion of this dataset specifically used to support the DeepQA component.

For DeepBench, the authors create a benchmark comprising ten deep learning tasks sourced from the Papers with Code (PWC) website.

The authors further introduce DeepLlama, which is Llama 3.1-8B fine-tuned on DeepData and outperforms the baseline models on DeepQA. Additionally, the authors develop a comprehensive evaluation pipeline, DeepBench, to evaluate and discuss the models' performance across 10 tasks before and after the performance of architectural improvement.

**Strengths:**

**Originality:**

As far as I know, DeepOnes is the first solution focused on high-level deep learning programs in this field and it also includes the capability for generating executable code. The authors discover the insufficiency of existing research on LLM agents in high-level deep learning program design and then contribute a comprehensive solution.

**Quality:**

1. The description of the evaluation pipeline in the paper is clear and easy to understand.

2. There are detailed descriptions and examples regarding the generation and composition of the dataset.

Overall, the quality of the paper is satisfactory.

**Significance:**

This work is a commendable attempt in both code generation and the LLM Agent field.

**Weaknesses:**

**Clarity:**

1. In the description of Figure 5, MA is mentioned, which does not appear in the figure; it might be more appropriate to refer to MD instead.

2. Some tasks have missing data in Table 2, particularly for speech recognition. The authors should clarify whether this is due to exceeding the execution time limit, failure in all iterations, or other reasons.

**Experiment:**

1. Lacking a discussion comparing the experimental results with the SOTA from Papers with Code, the study fails to demonstrate the advantages or limitations of the proposed solution relative to the SOTA across different tasks.

2. From Table 2, it can be observed that DeepLlama-8B underperforms Llama3.1-8B-Instruct on many tasks, which does not adequately demonstrate the improvement that DeepCode brings to the model. One possible reason is that, as the authors mention, GPT-4o lacks knowledge of the implementations of several recent techniques. This could provide valuable insights if the authors were to conduct a comparative analysis using a model that possesses this knowledge as the agent.

**Type Error:**

In line 438, 'execution' is incorrectly written as 'execuion'.

**Questions:**

**About Experiment:**

In Section 5.2, the training time limit is uniformly applied across different tasks, which may lead to the following two issues:
- Different models may require varying amounts of training time to converge on the same task while setting the same time limit could undermine the performance of models that are inherently expected to perform better.
- The model architectures for different tasks are usually different, resulting in distinct time requirements; thus, the time limits might be set specifically for each task.

The authors could consider task-specific time limits or provide justification for using a uniform limit to ensure the validity of experimental results.

Additionally, if my concerns about the experiments in the Weaknesses section are valid, it would be beneficial for the authors to enhance the clarity and comprehensiveness of the experimental section. If these concerns are not valid, please provide a detailed explanation.

---

### Official Review · Reviewer_YVsd · 2024-11-03

**Soundness:** 2
**Presentation:** 2
**Contribution:** 2
**Rating:** 5
**Confidence:** 3

**Summary:**

The authors introduce the task of Deep Learning Program Design (DLPD), which requires an agent to define a model to be trained for the given task. This includes designing the neural architecture and suggesting values for the hyperparameters.

The paper then presents 3 datasets (referred to as DeepOne), called DeepData, DeepQA and DeepBench. DeepData is a large dataset for finetuning LLMs on DPLD-specific knowledge. DeepQA is a small question-answering dataset for evaluating an LLM’s DLPD knowledge. DeepBench is a benchmark for evaluating an agent’s solution generating capabilities.

Finally, the paper provides an empirical evaluation of popular LLMs, together with a LLM, finetuned on DeepData.

**Strengths:**

The new task of DLPD appears potentially useful, as it brings attention to using LLMs for more high-level deep learning tasks.
The DeepOnes could be useful to the community.
The authors identify some of the limitations of their work.

**Weaknesses:**

Lack of precise definition of DLPD. Abstract and introduction say “designing an effective deep learning program for the task utilizing ...”. From this, I know what the program utilizes but I don’t know what it looks like, nor what the program space looks like. An example would be very beneficial.

(Arguably) limited novelty. The paper processes text from arxiv papers and their code, using pre-trained LLMs, but it’s not clear to me that any of these techniques represent novel ideas.

The results (Table 2) do not appear to convincingly demonstrate the advantages of the proposed ideas. For instance, finetuning LLama on DeepData does not appear to consistently outperform LLama-Instruct. Also, Claude 3.5 Sonnet+ appears to perform much better (e.g. on Face recognition: 3.49 vs 38.36). Moreover, the proposed method of “iterative suggestion” of the architecture doesn’t seem to consistently improve the performance.

**Questions:**

q1: You ask the agent to suggest improvements to the initially proposed architecture, without providing the agent with feedback the initial architecture’s performance. Why do you imagine this will work?

q2: Regarding Table 2: what’s the reason of Llama-Instruct having higher numbers on many of the rows, compared to DeepLLama?

q3: Overall, how would you say the experiments provided in the paper support the usefulness of DeepData?

---

### Note · Authors · 2024-11-21

I have read and agree with the venue's withdrawal policy on behalf of myself and my co-authors.